# Portable-CELLxGENE: standalone executables of CELLxGENE for easy installation

George T. Hall[1],[*]

**1** UCL Great Ormond Street Institute of Child Health, University College London, 20 Guilford Street, WC1N 1DZ, London, UK

## ABSTRACT

Biologists who want to analyse their single-cell transcriptomics dataset must install and use specialist software via the command line. This is often impractical for non-bioinformaticians. Whilst the popular CELLxGENE software provides an intuitive graphical interface to facilitate analysis outside the command line, its server-side installation and execution remain complex. A version that is easier to install and run would allow non-bioinformaticians to take advantage of this valuable tool without needing to use the command line. This work introduces Portable-CELLxGENE, a standalone distribution of CELLxGENE that can be installed via a graphical interface. It contains an easy-to-use extension of the CELLxGENE-Gateway Python package to allow the analysis of multiple datasets. This tool enables non-bioinformaticians to carry out simple analyses independently.

**Availability and implementation:** Versions of Portable-CELLxGENE for Windows and MacOS, along with source code, are available at https://george-hall-ucl.github.io/Portable-CELLxGENE-Docs. It is licensed under the GNU General Public License v3.

**Subjects** Software and Workflows, Bioinformatics, Biomedical Science

**Submitted:** 07 November 2024

**\*** E-mail: george.hall@ucl.ac.uk

Preprint submitted at https://doi.org/10.48550/arXiv.2408.11844

## STATEMENT OF NEED

Software to analyse single-cell transcriptomics experiments requires time and expertise to install and operate. Non-bioinformaticians must therefore often work in close collaboration with bioinformaticians to carry out tasks such as labelling cell types according to marker genes and assessing differential gene expression. Research would be accelerated if non-bioinformaticians could tackle these tasks alone. However, this is currently challenging due to the lack of easy-to-install and easy-to-use software.

The CELLxGENE tool [1] is easy-to-use since it provides a graphical user interface (GUI) through which single-cell datasets can be analysed. The CELLxGENE-Gateway extension [2] allows it to be used with multiple datasets, with the ability to create multiple versions of cell-level annotations and gene-level gene sets. However, both tools remain prohibitively complex for non-bioinformaticians to install since they require Python and the correct versions of many Python packages and other software to be installed. Whilst CELLxGENE instances can be hosted online – thereby bypassing installation for the end user – setting up the server requires time, money, and specialist knowledge, in addition to the requirement to remain online whilst conducting analysis.

Some other tools also provide a GUI for the analysis of single-cell transcriptomics experiments, but each has shortcomings. The Galaxy tool [3] provides a GUI and allows for

more complex analyses than CELLxGENE, but this complexity may make it unsuitable for non-specialists. Whilst the Loupe browser developed by 10× Genomics is easy to install and use, it can only operate on datasets saved in the proprietary `loupe` format. These datasets are generally those that have been processed by the 10× Genomics CellRanger software [4], although datasets in other formats can now be converted to this format [5]. Nevertheless, the Loupe browser is not open source, potentially limiting its future availability, extendibility, and utility in some situations. Portable-CELLxGENE, on the other hand, is build on open-source software using the well-maintained and widely used `h5ad` file format.

Portable-CELLxGENE addresses the lack of an easy-to-install and easy-to-use tool by providing standalone executable versions of the intuitive and powerful CELLxGENE software. It will thus accelerate research by enabling non-bioinformaticians to carry out simple analysis tasks independently from bioinformaticians.

## IMPLEMENTATION

### Portable-CELLxGENE

Portable-CELLxGENE comprises a conda [6] environment (containing Python, all necessary Python packages, and all other software) along with a script to run the tool. The conda environment incorporates CELLxGENE along with CELLxGENE-Gateway, an extension that allows for the analysis and annotation of multiple datasets. Upon launch, a shell is started that activates the conda environment. A window appears for the user to select the location of their dataset. The launch script then sets environment variables to point to the dataset and configure other CELLxGENE and CELLxGENE-Gateway options. Finally, a CELLxGENE-Gateway session is started, and, after a small delay to allow for all necessary processes to start, a browser window is opened to display the web app.

The homepage of the app is a slightly modified version of the CELLxGENE-Gateway file browser, designed to improve usability (Figure 1). From this page, datasets can be loaded for analysis in CELLxGENE. Sets of annotations and gene sets can be created or loaded, along with compiled Python/R notebooks showing how the data has been processed previously. Running CELLxGENE sessions can be identified and terminated if desired. Once a dataset has been loaded through this page, a CELLxGENE instance is launched and can be used in the standard way, with gene sets and cell annotations being created and saved as `csv` files (Figure 2). No internet connection is required since all software is hosted locally.

### Installation

Portable-CELLxGENE can be installed with a straightforward graphical interface on MacOS and Windows. On MacOS, a standard `dmg` installer allows the user to simply drag the application into the `Applications` directory (Figure 3). On Windows, an executable can be run that downloads and installs the software automatically (Figure 4).

### Uninstallation

Portable-CELLxGENE is similarly easy to uninstall. On MacOS, the application can simply be deleted from the `Applications` directory. On Windows, it can be deleted from the `Local Application Data` where it has been installed, along with its desktop and start menu shortcuts. A script for this process is available in the documentation.





**Figure 1.** Homepage of Portable-CELLxGENE displaying the user's datasets, annotations, and compiled analysis notebooks.

## Build process

The build process of Portable-CELLxGENE is designed to be straightforward for both MacOS and Windows. It is detailed in the documentation.

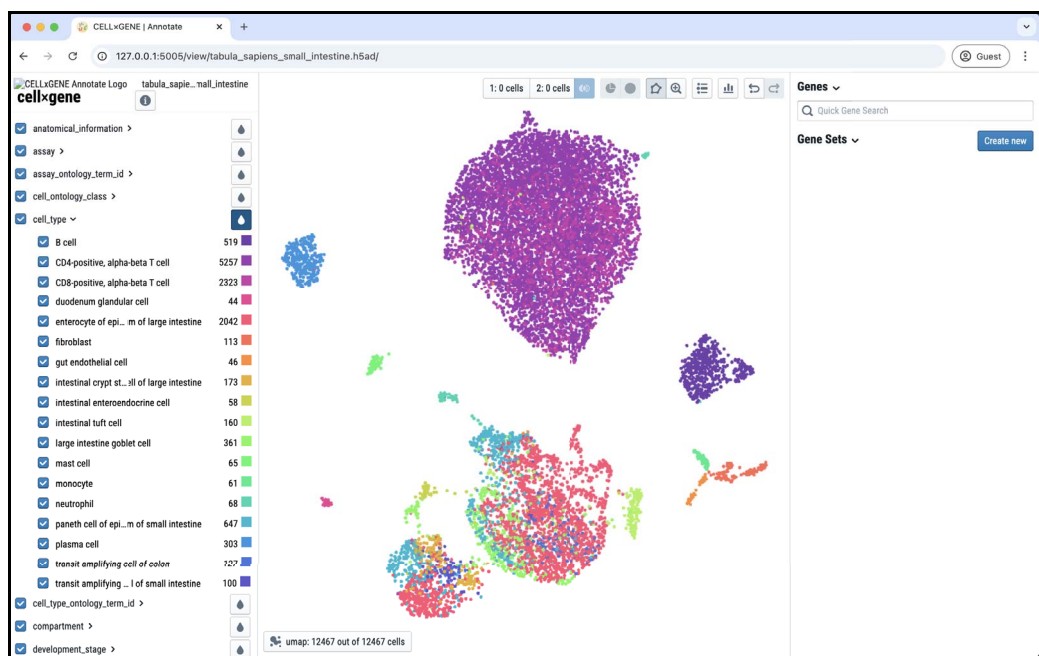

**Figure 2.** Screenshot of the CELLxGENE interface available within Portable-CELLxGENE.

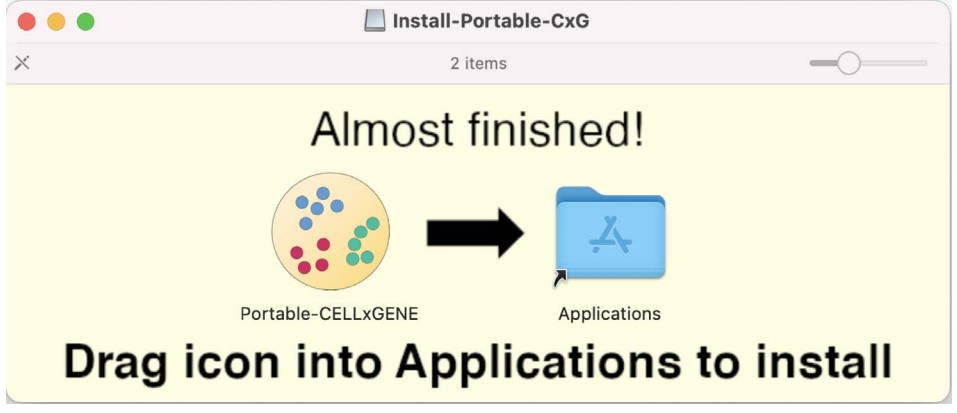

**Figure 3.** Screenshot of the MacOS `dmg` installer. The user simply drags the app icon into the `Applications` folder.

### *MacOS*

Bash scripts to automate the build process are available in the Portable-CELLxGENE GitHub repository. Briefly, the process comprises two stages: building and signing. In the building stage, a template `.app` directory (created using Platypus [7]) is loaded with the conda environment and a script to launch the program. In the signing stage, the MacOS code signing tool `codesign` is used to add an accredited developer identity to the app and all libraries and executables within the conda environment. Finally, `node-appdmg` [8] incorporates the signed software into a `dmg` file for easy installation, which itself is then signed and released.

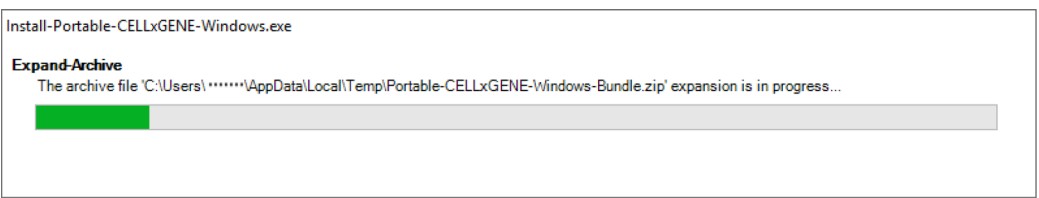

**Figure 4.** Screenshot of the Windows installer. The user downloads and runs a small executable, which installs the software.

### Windows

Build scripts, assets, and the Windows conda environment are all hosted on the `PortableCELLxGENE-assets` GitHub repository [9]. The batch script to activate the conda environment and run CELLxGENE-Gateway is converted into an executable with the `bat2exe` tool [10]. This executable can then be signed with an extended validation certificate. Finally, an installer is created by converting the powershell installation script downloaded from the assets repository into an executable using `ps2exe` [11]. This installer executable can then be signed and released.

## AVAILABILITY OF SUPPORTING SOURCE CODE AND REQUIREMENTS

- Project name: Portable-CELLxGENE
- Project home page: https://george-hall-ucl.github.io/Portable-CELLxGENE-Docs [12].
- Source code: https://github.com/george-hall-ucl/portable-cellxgene [13]
- Operating system(s): MacOS and Windows
- Programming language: Python, Bash, Batch, Powershell
- Other requirements: None
- License: GNU General Public License v3
- RRID:SCR_026140.

## CONTRIBUTING GUIDELINES

Developers are welcome to contribute to Portable-CELLxGENE by submitting a pull request on the GitHub repository [14]. All contributors are expected to adhere to the Code of Conduct [15].

## DATA AVAILABILITY

Code snapshots are available in Software Heritage [13].

Any dataset in the `h5ad` format can be used in Portable-CELLxGENE. CELLxGENE Discover [16] is one example of a database of files in this format. The `h5ad` file must be stored in a directory, which is then accessed with the folder selection window. More detailed instructions are available in the documentation.

Three datasets from the Tabula Sapiens [17] were downloaded from CELLxGENE Discover and used to generate the figures in this paper: `Large_Intestine` [18], `Small_Intestine` [19], and `Pancreas` [20]. Tabula Sapiens single-cell data is also available from Figshare [21].

## ABBREVIATIONS

GUI, graphical user interface.



## DECLARATIONS

### Ethical approval

Not applicable.

### Competing interests

None declared.

### Funding

The author is funded by the NIHR Great Ormond Street Biomedical Research Centre.

### Acknowledgements

Thank you to Joe Davidson, Yara Sanchez Corrales and others for testing and providing feedback on the installation process and design of Portable-CELLxGENE. Thank you to Sergi Castellano for feedback on the manuscript and support throughout the project. Thank you to the reviewers for insightful comments which improved the quality of the software and manuscript.

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
