## [Editor Report]

Editor’s AssessmentCELLxGENE is a popular open-source tool to find, download, and visually explore curated and standardized single-cell biology datasets. Biologists without the bioinformatics skills to use the command line but still want to analyse their single-cell transcriptomics datasets using this platform as it provides an intuitive graphical interface. Though its server-side installation and execution remain too complex and impractical for many. This paper presents Portable-CELLxGENE, standalone small executables of CELLxGENE for more easy installation. Providing downloadable versions of Portable-CELLxGENE for Windows and MacOS, along with its source code, that are available at GitHub https://github.com/george-hall-ucl/portable-cellxgene under a GPLv3 license. Peer-review helped streamline the MacOS installation and improve the documentation, and testing proved it was very easy to install and run. Test datasets are provided and developers are encouraged to contribute to Portable-CELLxGENE by submitting a pull request on the GitHub repository.Editor’s AssessmentCELLxGENE is a popular open-source tool to find, download, and visually explore curated and standardized single-cell biology datasets. Biologists without the bioinformatics skills to use the command line but still want to analyse their single-cell transcriptomics datasets using this platform as it provides an intuitive graphical interface. Though its server-side installation and execution remain too complex and impractical for many. This paper presents Portable-CELLxGENE, standalone small executables of CELLxGENE for more easy installation. Providing downloadable versions of Portable-CELLxGENE for Windows and MacOS, along with its source code, that are available at GitHub https://github.com/george-hall-ucl/portable-cellxgene under a GPLv3 license. Peer-review helped streamline the MacOS installation and improve the documentation, and testing proved it was very easy to install and run. Test datasets are provided and developers are encouraged to contribute to Portable-CELLxGENE by submitting a pull request on the GitHub repository.

---

## [Reviewer Report]

Reviewer name and names of any other individual's who aided in reviewerSidney M. Bell, PhDDo you understand and agree to our policy of having open and named reviews, and having your review included with the published manuscript. (If no, please inform the editor that you cannot review this manuscript.)YesIs the language of sufficient quality?YesPlease add additional comments on language quality to clarify if neededIs there a clear statement of need explaining what problems the software is designed to solve and who the target audience is? YesAdditional CommentsIs the source code available, and has an appropriate Open Source Initiative license <a href="https://opensource.org/licenses" target="_blank">(https://opensource.org/licenses)</a> been assigned to the code?YesAdditional Comments[SUGGESTION] While not a requirement, I would strongly encourage a more permissive license to promote further reuse and development of CELLxGENE and related packages. The GPL license used can have unintended consequences (for example, at CZI we build almost exclusively with a very permissive MIT license per institutional policy; this sometimes limits our ability to collaborate with other OSS projects when their GPL license would require that we also use this more restrictive license, even though we are building also openly).As Open Source Software are there guidelines on how to contribute, report issues or seek support on the code?YesAdditional CommentsIs the code executable?YesAdditional Comments[CONSIDERATION] This packages does include an override to allow unsigned executables; I do not have the security engineering background required to assess this, but I would advise the editor to seek their input.Is installation/deployment sufficiently outlined in the paper and documentation, and does it proceed as outlined?YesAdditional Comments[REVISION NEEDED] The UI is very confusing as it says to select (or drag/drop) a *file*, but it only accepts a *directory*. Please update this to be more clear; it almost prevented me from using the software entirely. I was only able to verify on Mac. Given that many experimental biologists rely on Windows, it would be ideal if another reviewer was able to verify on Windows.Is the documentation provided clear and user friendly?YesAdditional Comments[SUGGESTION] The documentation and videos in the readme are very excellent. This is not required and would be above-and-beyond, but the author may wish to consider a github pages site as this would be more approachable to non-computational users (github can be a bit intimidating) and perhaps also help with SEO and discoverability.Is there enough clear information in the documentation to install, run and test this tool, including information on where to seek help if required?YesAdditional CommentsIs there a clearly-stated list of dependencies, and is the core functionality of the software documented to a satisfactory level?YesAdditional Comments[CONSIDERATION] My only minor concern with this submission is the long-term maintenance incurred by the choice to fork the CELLxGENE-gateway. Has the author considered sending their changes upstream as a PR, instead of maintaining a fork? The UI improvements are nice, but I do wonder about the tradeoff in long-term maintenance as the Novartis team is well staffed and occasionally updates the gateway (e.g., there are changes to the environment variables in the Novartis version of the gateway from 10 months ago which have not been updated in this package's fork of the gateway). Similarly, the version of cellxgene is not pinned (either to a specific version or to `latest`) in either package's requirements; while there are advantages to the current unpinned approach, it is a minor risk worth pointing out. All of this said, however, this approach is reasonable and has its benefits. Neither CELLxGENE Explorer nor Gateway are rapidly being developed, and so the risks described above are probably minimal, assuming that the author has a plan to maintain the package occasionally as the need arises due to upstream changes.Have any claims of performance been sufficiently tested and compared to other commonly-used packages? Not applicableAdditional CommentsIs test data available, either included with the submission or openly available via cited third party sources (e.g. accession numbers, data DOIs)?YesAdditional CommentsAre there (ideally real world) examples demonstrating use of the software? YesAdditional CommentsIs automated testing used or are there manual steps described so that the functionality of the software can be verified?NoAdditional CommentsAny Additional Overall Comments to the AuthorThis is a fantastic extension of CELLxGENE's functionality which users have been asking for for years! Thank you for contributing to our community.RecommendationMinor Revisions

---

## [Reviewer Report]

Reviewer name and names of any other individual's who aided in reviewerAlexander KirchmairDo you understand and agree to our policy of having open and named reviews, and having your review included with the published manuscript. (If no, please inform the editor that you cannot review this manuscript.)YesIs the language of sufficient quality?YesPlease add additional comments on language quality to clarify if neededIs there a clear statement of need explaining what problems the software is designed to solve and who the target audience is? YesAdditional CommentsIs the source code available, and has an appropriate Open Source Initiative license <a href="https://opensource.org/licenses" target="_blank">(https://opensource.org/licenses)</a> been assigned to the code?YesAdditional CommentsAs Open Source Software are there guidelines on how to contribute, report issues or seek support on the code?YesAdditional CommentsIs the code executable?YesAdditional CommentsIs installation/deployment sufficiently outlined in the paper and documentation, and does it proceed as outlined?YesAdditional CommentsIs the documentation provided clear and user friendly?YesAdditional CommentsIs there enough clear information in the documentation to install, run and test this tool, including information on where to seek help if required?YesAdditional CommentsIs there a clearly-stated list of dependencies, and is the core functionality of the software documented to a satisfactory level?YesAdditional CommentsHave any claims of performance been sufficiently tested and compared to other commonly-used packages? Not applicableAdditional CommentsIs test data available, either included with the submission or openly available via cited third party sources (e.g. accession numbers, data DOIs)?YesAdditional CommentsA more detailed example on how to download test data and use them with Portable-CELLxGENE could be included in the documentation.Are there (ideally real world) examples demonstrating use of the software? YesAdditional CommentsIs automated testing used or are there manual steps described so that the functionality of the software can be verified?NoAdditional CommentsNot applicableAny Additional Overall Comments to the AuthorPortable-CELLxGENE is a practical solution to lower the accessibility of CELLxGENE for non-bioinformaticians. I tested it on several datasets, and it worked as expected. While the tool does not introduce any novel methods, its easy installation and GUI-based workflow can make it a useful contribution to the single-cell community.RecommendationAccept